# *Homo sapiens*: The Superspreader of Plant Viral Diseases

**DOI:** 10.3390/v12121462

**Published:** 2020-12-17

**Authors:** Buddhini Ranawaka, Satomi Hayashi, Peter M. Waterhouse, Felipe F. de Felippes

**Affiliations:** Centre for Agriculture and the Bioeconomy, Institute for Future Environments, Queensland University of Technology (QUT), 2 George Street, Brisbane, QLD 4000, Australia; ra.ranawaka@qut.edu.au (B.R.); peter.waterhouse@qut.edu.au (P.M.W.)

**Keywords:** plant viruses, viral vectors, plant diseases, virus spread

## Abstract

Plant viruses are commonly vectored by flying or crawling animals, such as aphids and beetles, and cause serious losses in major agricultural and horticultural crops. Controlling virus spread is often achieved by minimizing a crop’s exposure to the vector, or by reducing vector numbers with compounds such as insecticides. A major, but less obvious, factor not controlled by these measures is *Homo sapiens*. Here, we discuss the inconvenient truth of how humans have become superspreaders of plant viruses on both a local and a global scale.

## 1. Introduction

In the year 2020, the world has seen the fast and perverse spread of SARS-CoV-2, which has led to a shutdown of our societies and the loss of 1.3 million human lives worldwide so far [1]. Although unfamiliar to most people, plants are also susceptible to infection by wide range of viruses. Furthermore, damages caused by plant viruses on human lives can be as strong, or even more serious than those caused by their animal counterparts. Throughout history, the outbreak of diseases caused by plant viruses have been major contributors to chronic food insecurity [2], a scenario that tends to worsen with our ever-growing population.

Plant viruses constitute a major cause of plant diseases with an estimated economic impact of more than USD 30 billion annually [3]. Some viruses can wipe out entire plantations, resulting in 100% yield loss [4,5] and, subsequently affecting the revenue of farmers, increasing the price of food, and in more extreme cases, its availability to the market. Globally, the most destructive plant viruses are identified to be members of begomoviruses, tospoviruses and potyviruses. Significant epidemics caused by these viruses include not only those affecting economically important plants, but also staple food crops such as cassava, maize, rice and banana. Therefore, in addition to causing damage to farmers’ and countries’ economies, such plant disease epidemics can also lead to the starvation of a significant portion of the world’s population who depend on these plants for their subsistence [3,6].

While the spread of animal viruses is most often associated with direct contact or proximity to infected individuals, plant viruses are transmitted through wounds on the plant or via a vector, most often insects, fungi and nematodes that feed or infect the plant [3]. Although these vectors have often been the major target for controlling the spread of plant viral diseases, it is apparent that human activities also play a major role in the dissemination of plant viruses (Table 1). Man has distributed most of the cultivated plants around the world by removing them from their centre of domestication. As such, humans are greatly responsible for the novel encounters between plants and their pests [7]. Since many plant viruses have a broad range of hosts and vectors [8], introduction of crops to a new area can enable indigenous viruses from native plants to spread to the crops, and vice versa [3]. Moreover, modern agricultural systems, such as monocultures, have intensified and altered agricultural practices. Continuous cropping patterns encourage the accumulation of viruses and proliferation of their vectors in the field, leading to pandemics.

## 2. Direct Human Intervention in Virus Spread

There are several ways in which humans currently affect the spread of plant viral diseases. For example, the exchange of virus contaminated material between people plays a major role in transferring the virus to uninfected plants, most often as a result of limited knowledge in viral aetiology of symptomatic plants. For instance, the initial course for the spread of both African cassava mosaic virus (ACMV) [11] and sweet potato virus disease (SPVD) [30] is the exchange of infected stem cuttings and vines, respectively. If farmers are not vigilant, purchasing plant materials (i.e., seeds and tissues for vegetative propagation) from uncertified seed networks can increase the risk of global dissemination of plant virus diseases [31]. The effects are the same with the use of infected plant material for grafting, budding, and transplanting [17].

Another common way in which some viruses spread within crop fields is due to poor agricultural practices, such as the usage of unsterilised tools, not clearing plant debris, and even the continuous use of clothes and shoes that have been in the contaminated field [30,32]. No-till farming is a technique with several benefits to agriculture. However, not removing plant material from one season to another in contaminated fields can spread the virus to new plants and increase its accumulation [30]. Tobacco mosaic virus (TMV) is the typical case where the spread of the disease benefits from continuous cropping system, as it can survive or hibernate in crop debris, soil and other perennial hosts. In addition, these viruses can transmit within the field through mechanical wounds caused by contaminated tools, clothes, and footwear [26]. Interestingly, TMV is also capable of spreading via tobacco products (i.e., air-cured tobacco), where smokers rolling their cigarettes can transmit the virus with their contaminated hands [27].

Some plant diseases rely heavily on insect vectors for the transmission of the virus to a healthy plant. Tomato yellow leaf curl disease is one such case where the disease spreads by the feeding of whitefly vector carrying tomato yellow leaf curl virus (TYLCV) [7]. In this specific example, the insect-mediated viral spread is limited by the flight range of the whiteflies [33]. However, long-distance movement of insect-infested material/commodities by humans have tremendous consequences to how far the insect vector, and therefore the disease, can spread. Indeed, accidental import/export of insect vector-contaminated materials are identified as a major cause of plant virus outbreaks [12].

## 3. Virus Spread Coupled with Climate Change

The successful emergence and spread of plant viruses, and that of their vectors, are also indirectly influenced by the behaviour of mankind. Global climate change linked to human activities has increased global temperature and CO_2_ concentrations, leading to altered rainfall patterns, recurrent extreme weather events, as well as variations in wind velocity and direction [3,34,35]. Such changes have a range of impacts on the host plants, the virus, and their vectors. While some of these events can be beneficial for the plant to fight against infections, an abrupt change in the climatic conditions can also be especially helpful for the dissemination of viral diseases [36]. For example, elevated temperatures have been shown to enhance small RNA mediated defence against ACMV and cymbidium ringspot virus in *Nicotiana benthamiana* [37,38]; however, it also increases the contact transmission, the rate of virus multiplication and systemic movement of the virus within the plant [39]. In addition, higher temperatures are favourable for insects as vectors due to the increase in numbers of winged aphid morphs [40], shorter adult-to-adult generation time [41] and increased flight activity [42]. Moreover, alterations in wind speed and direction can affect how viruliferous vectors disseminate over long distances, affecting their distribution [39].

## 4. Challenges in Mitigating Plant Viral Diseases

Undoubtedly, lifestyle and reluctance to heed science-based information, at both an individual and societal level, have been major reasons contributing to the current COVID-19 pandemic. Modern people are accustomed to frequent domestic and international travel, large gatherings such as sporting events and concerts, all of which have played a central role in how fast and far the virus has spread. Similarly, the transmission of exotic plant viruses across local and international borders has been aggravated along with increased global trades of food and agriculture products. In addition, food items infected with viruses can easily travel across borders with the world’s population travelling more often and further. Overall, trade-in plants, plant products, and the movement of people are accountable for the 71% of factors known as routes of emerging plant viral diseases, while 16% is due to change in the vector populations [43]. A few examples of viruses intercepted at Australian and New Zealand quarantine stations, where strict quarantine measurements are in place, are peanut stripe virus G, apple stem grooving virus, grapevine virus B and sweet potato virus G [43].

The COVID-19 pandemic has shown us the importance of containment measures, such as self-isolation and quarantine, in halting the spread of the disease [44]. The same strategy can also be applied to combat the spread of plant viruses. Indeed, the spread of banana bunchy top virus, potato leafroll virus, sugarcane mosaic virus and plum pox virus have been controlled using effective containment programmes [43] (Figure 1). However, such approaches are limited to situations where there are reliable diagnostics, appropriate infrastructure and community adherence to regulatory protocols. This method is heavily dependent on the commitment and actions of local and federal governments, as well as individuals, which is not always the case. It seems unlikely that the extreme actions leading to changes in our lifestyle, as seen for the COVID-19 crisis, can be easily implemented for fighting against plant viruses.

## 5. Conclusions

Ultimately, unless the threat of virus infection of food crops is perceived to be of sufficient impact (as may one day be the case due to the escalating world population and reducing areas of fertile arable land), changing human behaviour in order to minimise crop losses seems less likely to be achieved than the development of crops with new sources of virus or vector resistance. To finish on an optimistic note: never before has humanity possessed such extensive genomic information and insights about crops, their wild relatives, their pathogens and their pests; nor has it possessed such powerful molecular and genetic technologies for accelerated breeding and synthetic biology. It is probably with this information and these tools that resilient crops can be developed to increase sustainable food supplies to such a level that they offset the damages wrought by *Homo sapiens,* the superspreader of plant virus diseases.

## Figures and Tables

**Figure 1 viruses-12-01462-f001:**
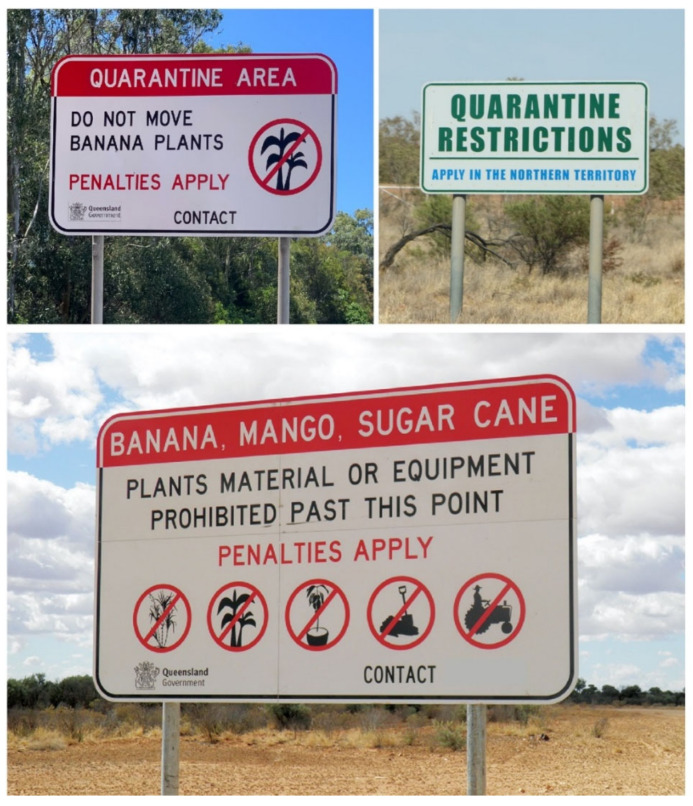
Containment measures as a strategy to mitigate the spread of plant viral diseases. Some countries such as Australia have strong policies to halt the spread of plant diseases, including viral ones, based on confinement and limitation on the movement of plant material and equipment. Image credit (bottom picture): Biosecurity Queensland.

**Table 1 viruses-12-01462-t001:** Human intervention in the spread of plant viruses.

Virus	Genus	Vector	Host	Human Intervention
Banana bunchy top virus (BBTV)	Babuvirus	Aphids	Banana, plantain, abaca and other plants in family Musaceae	Trade of vegetative planting and tissue culture material [9]
Banana streak virus (BSV)	Badnavirus	Mealybugs	Banana, *Heliconia*	Trade of vegetative planting and tissue culture material [10]
African cassava mosaic virus (ACMV)	Begomovirus	Whitefly	Cassava, castor bean	Exchange of virus infected plant material. Trade of infected seeds and plant material [11]
Tomato yellow leaf curl virus (TYLCV)	Begomovirus	Whitefly	French bean, Solanaceous plants	Accidental movement of insect vector [12]
Cauliflower mosaic virus (CaMV)	Caulimovirus	Aphids	Cauliflower, Chinese cabbage, brussels sprout, turnip	Virus contaminated machinery, equipment and workers [13].Trade of infected plant material [14]
Cucumber mosaic virus (CMV)	Cucumovirus	Aphids	Soybean, tobacco, pepper	Trade of infected seeds and plant material [15]
Tobacco necrosis virus (TNV)	Necrovirus	*Olpidium brassicae*	French bean, cowpea, mung bean, melon, tulip, tobacco, cucumber	Virus contaminated machinery, equipment and workers [16]
Plum pox virus (PPV)	Potyvirus	Aphids	Apricots, peaches, plums, almonds	Grafting, budding, and transplanting of infected plant material [17]
Potato virus Y (PVY)	Potyvirus	Aphids	Potato, tomato, capsicum, tobacco	Virus contaminated machinery, equipment and workers [18]. Trade of infected plant material [19]
Maize dwarf mosaic virus (MDMV)	Potyvirus	Aphids	Maize, sugarcane, sorghum	Trade of infected plant material [20]
Sweet potato feathery mottle virus (SPFMV)	Potyvirus	Aphids	Sweet potato, wild *Ipomoea* sp. *Nicotiana* sp.	Trade of infected tubers and cuttings, grafting, and mechanical inoculation [21]
Zucchini yellow mosaic virus (ZYMV)	Potyvirus	Aphids	Cucumber, pumpkin, rockmelon, zucchini	Virus contaminated machinery, equipment and workers [22]
Sugarcane mosaic virus (SCMV)	Potyvirus	Aphids	Sugarcane and Poaceae plants	Trade of infected plant material [23]
Rice yellow mottle virus (RYMV)	Sobemovirus	Adult beetles in family Chrysomelidae, Grasshoppers	Rice	Virus contaminated sickles, hands and crop residues. Tight contact between plants during planting [24,25]
Tobacco mosaic virus (TMV)	Tobamovirus	Grasshoppers, Caterpillars (occasionally)	Tobacco, tomato, Solanaceous plants	Virus contaminated machinery, equipment, farmers and smokers [26,27]
Tomato spotted wilt virus (TSWV)	Orthotospovirus	Thrips	Peanut, pepper, tomato	Use of infected cuttings and bulbs for propagation [28]
Rice tungro virus (RTV)	Tungrovirus	Leafhopper	Rice	Trade of infected seeds, and transplanting infected seedlings [29]

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
