# Peer review of "Homo sapiens: The Superspreader of Plant Viral Diseases"

_viruses, 2020, doi:10.3390/v12121462_

Round 1
Reviewer 1 Report
Manuscript viruses-1027019 is an interesting opinion piece on the active role of humans in the dissemination of plant viruses at a local and global scale. Using a limited number of examples from the literature, authors make a strong case for the direct involvement of humans in the movement of viruses. Given the current COVID-19 pandemic, the paper is timely and interesting. In addition, the manuscript is overall well written, although some minor changes should be considered for improvement. See recommendations below:
Line 17: Uncontrollable should be toned down, as several countries have clearly been very successful at controlling SARS-CoV-2. Maybe perverse would be a better term.
Line 18: ... 1.3 million human lives worldwide so far [1].
Line 26: Plant viruses do not affect the productivity of farmers; they affect the productivity of crops
Line 27: to the market. Globally, the most destructive ...
Line 35: ... these vectors have often been ...
Line 37: What is a modern man? Is a human that lived more than 10,000 years a modern man? Maybe humans would be more appropriate.
Line 45, Table 1, column Virus: Add a space between virus and (SPFMV)
Line 45, Table 1, column Genus: Change Tospovirus to Orthotospovirus
Page 3, line 24: ... whitefly vector carrying tomato yellow leaf curl virus (TYLCV) [7].
Page 3, line 37: ... have been shown to enhance small RNA ...
Page 3, line 38: Change Cymbidium to cymbidium
Page 3, line 45: Change ignorance to lack of desire to adopt science-based information
Page 4, line 3: ... travel across borders with the ...
Page 4, line 7: Change Peanut to peanut
Page 4, line 7: Change Apple to apple
Page 4, line 8: Change Grapevine to grapevine
Page 4, line 8: There are more than 90 viruses identified in grapevines. Which one(s) are you referring to here?
Page 4, line 8: Change Sweet to sweet
Page 4, line 14: ... adherence to regulatory protocols. This ...
Page 4, line 15: ... actions of local and federal governments, as well as individuals, which is not ...
Page 5, line 3: Change made to developed
Page 5, line 4: ... the major plant virus ...
Reviewer 2 Report
The authors are free to discuss any own ideas. However, I would disagree with the authors when they make the main statement of paper, that is that humans can be considered as plant virus vectors.
In the current literature, plant virus vectors are organisms that DIRECTLY carry viruses in persistent or non-persistent manner. In this paper, the authors provide examples of human influence on spread of plant viruses on the Earth, however humans do not serve as virus carriers, as they do not carry virus particles in their bodies and just facilitate, in different ways but always indirectly, the spread of plant viruses. Therefore, humans are not plant virus vectors.
Thus, the main statement of this paper is not justified in the text.
Minor point:
References to the SARS-CoV-2 pandemic looks strange in the context of this paper (the first phrase of the Introduction and page 4). I suggest removing these phrases.